# Biofilm Production by Critical Antibiotic-Resistant Pathogens from an Equine Wound

**DOI:** 10.3390/ani13081342

**Published:** 2023-04-13

**Authors:** Ana C. Afonso, Mariana Sousa, Ana Rita Pinto, Mário Cotovio, Manuel Simões, Maria José Saavedra

**Affiliations:** 1LEPABE—Laboratory for Process Engineering, Environment, Biotechnology and Energy, Faculty of Engineering, Department of Chemical Engineering, University of Porto, 4200-465 Porto, Portugal; 2ALiCE—Associate Laboratory in Chemical Engineering, Faculty of Engineering, University of Porto, 4200-465 Porto, Portugal; 3CITAB—Centre for the Research and Technology of Agro-Environmental and Biological Sciences and Inov4Agro, University of Trás-os-Montes e Alto Douro, 5000-801 Vila Real, Portugal; 4Department of Veterinary Sciences-Antimicrobials, Biocides & Biofilms Unit, University of Trás-os-Montes and Alto Douro, 5000-801 Vila Real, Portugal; 5CECAV-Veterinary and Animal Research Centre and Associate Laboratory for Animal and Veterinary Science (AL4AnimalS), University of Trás-os-Montes and Alto Douro, 5000-801 Vila Real, Portugal

**Keywords:** antimicrobial resistance, biofilm, one health, *Pseudomonas aeruginosa*, *Staphylococcus aureus*, zoonotic disease

## Abstract

**Simple Summary:**

Antimicrobial resistance (AMR) is one of the biggest concerns of this century, threatening humans, animals, and the environment. Animals are important reservoirs of AMR, and contribute to its dissemination, since they are all interconnected. Through the microbiological analysis of a horse chronic wound, two bacterial species were isolated and identified as pathogens of high and critical priority to public health. Furthermore, these pathogens were able to form biofilms—the ability to organize themselves into communities—making them even more resistant to antibiotics. This study demonstrates the need for proper diagnosis and treatment of the chronic wounds of animals, as well as the urgent need to control this pandemic, which is AMR.

**Abstract:**

As in human medicine, in veterinary medicine, chronic wounds are often related to polymicrobial infections and the presence of a biofilm, which compromises the effectiveness of therapeutic approaches. In this study, a Lusitano mare presented a 21-day-old chronic wound that was only being treated with an antiseptic. A swab sample was collected, and three isolates of *Staphylococcus aureus* and one of *Pseudomonas aeruginosa* were isolated. *S. aureus* did not show resistance to a panel of antibiotics. However, the *P. aeruginosa* isolate showed a resistance profile to carbapenems and fluoroquinolones, which may suggest a cross-resistance between antiseptic and antibiotics, given that no antibiotic therapy was applied to the wound or the mare in the previous year. Further experiments were conducted to assess the ability of the isolates to form biofilms, and to ascertain their susceptibility to gentamicin. The results demonstrated that the isolates produced biofilms. Gentamicin at the minimum inhibitory concentration (MIC) and 10× MIC caused biofilm removal between 59.3% and 85.7%, with the highest removal percentage being obtained for the *P. aeruginosa* isolate (at 10× MIC concentration). This study reveals that an equine wound was colonized by antibiotic resistant bacteria, and that all the wound colonizers could form biofilms, demonstrating the relevance of an adequate diagnosis and treatment when there is a suspicion of a biofilm-infected wound. It also highlights the possibility of resistance transmission between animals, animals and humans, or animals and the environment.

## 1. Introduction

Unfortunately, for horses, wounds constitute one of the most prevalent emergency conditions [1]. By themselves, horses already present a challenge in wound healing in the distal limbs [2,3], mainly due to the anatomical characteristics of this region where there is a substantial loss of soft tissue, high skin tension, and movement [4]. As if all the suffering and discomfort caused to the animal were not enough, affected horses lose their economic value and become unfit for work for long periods, or in other cases, their athletic careers end [5]. Equine wounds are associated with a high risk of infection due to the surrounding environment [6,7]. Infected wounds harbor diverse populations of microorganisms, making identification and treatment difficult, and resulting in chronic wounds that do not heal [8]. Equine and human chronic wounds are pathologically similar [9]. However, unlike human chronic wounds, the microbiology of equine wounds and biofilm infection has been inadequately studied [9,10,11]. In human wounds, chronicity has been attributed to the ability of bacteria to organize themselves into heterogenous communities, the so-called biofilms [12]. Biofilms are defined as complex communities consisting of microbial cells embedded in a self-produced matrix of extracellular polymeric substance (EPS) [13,14]. The presence of biofilms in human wounds has been found in approximately 60–80% of chronic wounds and 6% of acute wounds, which strongly implies their contribution to hindering wound healing [6,7]. However, biofilms do not only affect the ability to heal, but also play a protective role against stressful conditions [15]. A clear example is the protection against antimicrobials, which has already been shown to be 10–1000 times higher compared to planktonic cells [16]. Another advantage of biofilms, over the individual members that compose them, is the fact that they harbor several types of microorganisms, including pathogens, and others carrying resistance determinants [13].

On the other hand, the presence of biofilms and/or multiresistant microorganisms in animals not only complicates their clinical outcomes, but also poses a risk to other animals and their caregivers. In recent years, the relationship between domestic animals and their owners has attracted attention regarding the acquisition and transfer of genes that confer resistance to antibiotics [17,18]. In fact, research on the epidemiology and transmission of resistant bacteria between humans and animals and vice versa has increased, and its zoonotic potential cannot be neglected [19,20].

In this study, four bacterial isolates from a mare’s infected wound were identified and characterized in terms of their antibiotic susceptibility profile and biofilm formation ability. In addition, single-species biofilms were exposed to gentamicin, and the effectiveness of this antimicrobial in biofilm removal was assessed.

## 2. Detailed Case Description

### 2.1. Sample Collection and Processing

Samples were obtained from a 21-day chronic wound located on the lateral aspect of the cannon bone from the left hind limb of a 12-year-old Lusitano mare used for recreation. The wound had been treated locally with chlorhexidine, but without local or systemic antibiotic treatment. For the collection of microbiological samples, the wound was washed with a sterile saline solution to remove surface debris. Then, a surgical swab was taken from the surface of the wound and stored in Stuart’s transport medium (Oxoid, Hampshire, UK), and sent to the Medical Microbiology Laboratory—Antimicrobials, Biocides, and Biofilms Unit, Department of Veterinary Sciences, UTAD. The entire procedure was conducted in accordance with the European Animal Welfare Directives (Directive 98/58/CE and Decreto-lei 64/2000).

Samples were cultured in Brain Heart Infusion (BHI, Oxoid, Hampshire, UK) broth and incubated at 37 ± 2 °C, for 24 h. After this period, inocula with turbidity were considered positive for bacterial growth, and selective and differential growth mediums [GSP *Pseudomonas Aeromonas* Selective Agar (Merck, Taufkirchen, Germany), Chromocult Coliform Agar (Oxoid, Hampshire, UK), MacConkey (Oxoid, Hampshire, UK), Baird-Parker Agar (Oxoid, Hampshire, UK) and Mannitol (Oxoid, Hampshire, UK)] were used for the isolation process.

### 2.2. Bacterial Isolates and Culture Conditions

After subculture, bacterial isolates were phenotypically identified through the automated Vitek^®^ 2 compact system (BioMerieux, Paris, France) using the Vitek^®^ 2 ID card for Gram-negative and Gram-positive (BioMérieux, Inc., Durham, NC, USA). The bacterial isolates were stored in cryovials with 30% (*v*/*v*) glycerol, at −80 ± 2 °C. For recovery, bacteria were cultured in MHA (Merck, Taufkirchen, Germany) plates and allowed to grow for 24 h at 37 ± 2 °C.

The inoculum needed for each assay was prepared by growing the isolates overnight in Mueller–Hinton Broth (MHB) (Merck, Taufkirchen, Germany), at 37 ± 2 °C with agitation (150 rpm). The optical density (OD) was adjusted to 10^8^ CFU/mL (OD_600nm_ of 0.132 for *Staphylococcus aureus* and OD_600nm_ of 0.333 for *Pseudomonas aeruginosa*). For this purpose, a spectrophotometer (Thermo Electron Corporation, EP1000T) was used.

### 2.3. Minimum Inhibitory Concentration (MIC) Determination

Minimum inhibitory concentrations (MICs) were obtained by a VITEK^®^ 2 compact system (BioMerieux, Paris, France). Automated antimicrobial susceptibility testing was performed using the Vitek^®^ 2 AST-GN97 card for Gram-negative, and the Vitek^®^ 2 AST-GP80 card for Gram-positive (BioMérieux, Inc., Durham, NC, USA), in accordance with the manufacturer’s specifications. All antibiotics tested for each isolate are listed in Table 1.

### 2.4. Biofilm Formation and Classification

The biofilm formation assay was performed according to the modified microtiter plate method described by Stepanović et al. [21]. Two plates, one for the 24 h analysis and one for the 48 h analysis, were prepared. Briefly, 200 µL of inoculum was added to each well. Control wells contained sterile fresh MHB without bacterial cells (200 μL). Then, microtiter plates were incubated aerobically at 37 ± 2 °C and 150 rpm. At 24 h of incubation, the medium of the 48 h plate was carefully discarded and replaced by fresh medium.

After biofilm development at 24 h and 48 h, the content of the wells was removed and washed with 200 μL of sterile saline solution (8.5 g/L NaCl) to discard nonadhered bacteria. The remaining attached bacteria were fixed with 250 µL of 99% (*v*/*v*) ethanol for 15 min. Then, the content of each well was discharged and the fixed bacteria were stained for 5 min with 200 μL of 1% (*v*/*v*) crystal violet (CV). The excess stain was gently withdrawn, and the dye bound to the adherent cells was solubilized with 200 µL of 33% (*v*/*v*) glacial acetic acid. The OD was measured at 570 nm using a microtiter plate reader (SPECTROstar^®^ Nano, BMG LABTECH).

Bacteria were classified using the scheme proposed by Stepanović et al. [21], where OD corresponds to the absorbance of wells containing cells, and ODc corresponds to the absorbance of wells containing only MHB: nonbiofilm producer: OD ≤ ODc; weak biofilm producer: ODc < OD ≤ 2 × ODc; moderate biofilm producer: 2 × ODc < OD ≤ 4 × ODc; strong biofilm producer: OD > 4 × ODc.

### 2.5. Biofilm Control

To determine whether gentamicin had any effect on biofilm control, 24 h old biofilms formed in 96-well microtiter plates were exposed to the MIC and 10× MIC of gentamicin, according to Simões et al. [22]. For that, after biofilm formation for 24 h, the content of each well was removed and washed with 200 µL of sterile NaCl solution. Then, 10 µL of gentamicin (Merck, Portugal) at the MIC and 10× MIC was applied in each well, and 190 μL of sterile fresh MHB was also added. Controls were composed of wells with sterile MHB and wells with bacterial suspension. The microtiter plates were incubated for 24 h at 37 ± 2 °C, and agitated at 150 rpm in the same refrigerated incubator with an orbital shaker. At 24 h after exposure, biofilms were analysed in terms of mass reduction by CV staining, as referred to in Section 2.4. At the end, the OD was measured at 570 nm using a microtiter plate reader (SPECTROstar^®^ Nano, BMG LABTECH) [23].

According to Monte et al. [24], the percentage of biomass reduction was calculated using the following equation, where, OD untreated_570nm_ is the OD of the biofilm mass at 24 h without any treatment, and OD treated_570nm_ is the OD of the biofilm mass at 24 h after antibiotic exposure.
$$\%\;\mathrm{of}\;\mathrm{Biomass}\;\mathrm{removal}=\frac{{\mathrm{OD}\;\mathrm{untreated}}_{570\mathrm{nm}}-{\;\mathrm{OD}\;\mathrm{treated}}_{570\mathrm{nm}}}{{\mathrm{OD}\;\mathrm{untreated}}_{570\mathrm{nm}}}\times 100$$

Biomass reduction was classified as follows—low efficacy: <25%; moderate efficacy: ≥25% to <60%; high efficacy: ≥60% to <90%; and excellent efficacy: ≥90% to ≤100% [25].

### 2.6. Statistical Analysis

The data were analysed using the statistical program SPSS version 27.0 (IBM, Statistical Package for the Social Sciences, New York, NY, USA). All data were statistically analysed using the paired Student’s *t*-test, based on a confidence level of 95% (*p* < 0.05 was considered statistically significant).

## 3. Results and Discussion

As described in Section 2.1, a 12-year-old Lusitano mare was assisted due to a 21-day chronic wound. The wound was treated with an antiseptic agent, but no antibiotic therapy was applied (nor in the previous year). From this wound, a microbiological sample was collected, which, after processing, was determined to be polymicrobial. Four different bacterial isolates were identified, three corresponding to *Staphylococcus aureus* and one to *Pseudomonas aeruginosa*. In fact, other studies have already shown that horse wounds often harbor several bacterial species [9,10]. Among the most isolated species are bacteria from the genus *Staphylococcus*, including *S. aureus* [9,10,26]. In addition, other species, such as *P. aeruginosa, Enterococcus faecium, E. faecalis, S. epidermidis* and *Serratia marcescens*, are commonly found [9,10]. Both *S. aureus* and *P. aeruginosa* species are included on the WHO list of antibiotic-resistant “priority pathogens” that pose the greatest threat to public health, as high and critical priority, respectively [27].

All isolates were identified and characterized in terms of their susceptibility profiles (Table 1). Despite the three *S. aureus* having a very similar susceptibility profile, the positive result in the cefoxitin screening test for *S. aureus* BPA2 should be highlighted. This test is recommended for the detection of methicillin resistance in *S. aureus* (MRSA), since cefoxitin is a potent inducer of the *mecA* regulatory system [28]. The *mecA* gene is a major determinant of resistance to β-lactam antibiotics, which encodes the novel penicillin-binding protein 2a (PBP 2a) [29]. Considering this isolate as resistant to β-lactams, the antibiotic therapy to be performed, according to the European Medicines Agency (EMA) for horses, would be one of the following antibiotics from Class D (use with prudence): trimethoprim/sulfamethoxazole, nitrofurantoin, tetracycline, or doxycycline. Only if these antibiotics are not clinically effective, should options from the above classes be used (Class C—use with caution; Class B—restricted use; Class A—avoid) [30]. This categorization groups antibiotics, considering both the risk that their use in animals entails for public health (due to the possible development of antimicrobial resistance), and the need for their use in veterinary medicine. For example, Class B antibiotics are strictly restricted for use in animals to mitigate the risk to public health. Class A, on the other hand, includes antibiotics that are not currently authorized in veterinary medicine in the European Union [30].

In the case of *P. aeruginosa,* it is resistant to carbapenem antibiotics (imipenem and ertapenem) and fluoroquinolones (enrofloxacin and marbofloxacin). Resistance to carbapenems is a serious and ongoing public health problem, as they are considered a last-resort treatment [31]. This fact presupposes resistance to other β-lactam antibiotics, resulting from intrinsic resistance or mediated transferable carbapenemase-encoding genes [31]. Furthermore, it has an intermediate susceptibility profile to the aminoglycoside gentamicin, which suggests that the effect will be dependent on the site of infection and the concentration used [30]. These results show that we are facing an isolate classified as a critical priority in terms of antibiotic resistance, *P. aeruginosa* MH/BHI; and, considering the positive result of the cefoxitin screening test, an isolate of *S. aureus* (BPA2) that fits the high priority profile. According to the EMA guidelines for horses, *P. aeruginosa* MH/BHI would be susceptible to the antibiotic amikacin from Class C (use with caution) [30]. However, a cross-resistance between biocides and antibiotics cannot be overlooked. In fact, several studies and with different strains have shown that subinhibitory concentrations of chlorhexidine induce resistance to chlorhexidine, and decrease susceptibility to antibiotics [32,33,34].

In addition to the presence of resistant bacteria in this animal wound, the presence of a biofilm would make healing from the infection even more difficult [35,36]. That said, all isolates were evaluated for their biofilm formation ability (Figure 1 and Table 2). The outcomes demonstrated that all bacteria were able to form biofilms after 24 h and 48 h of incubation.

Moreover, all isolates of *S. aureus* were classified as moderate biofilm producers, according to the classification of Stepanović et al. [21], for both 24 h and 48 h. However, for *S. aureus* Mannitol, there was a statistically significant decrease from 24 h to 48 h (*p* < 0.05). This can be explained by the intrinsic variability of the crystal violet (CV) assay, due to the removal of extracellular polymeric substances (EPS) when handling the microtiter plate [37]. For *S. aureus* BPA1, the highest biomass amounts were observed at 48 h, while for *S. aureus* BPA2 the highest biomass amounts were observed at 24 h. For both situations, the difference was not statistically significant (*p* > 0.05). *P. aeruginosa* MH/BHI was classified as a moderate biofilm producer at 24 h and as a strong biofilm producer at 48 h, with the highest production of biomass being observed at 48 h (*p* < 0.05).

To understand whether the presence of biofilm would influence the action of an antibiotic, single biofilms of each isolate were exposed to the antibiotic gentamicin at the MIC and 10× MIC. This antibiotic belongs to Class C, and was selected considering its current use in both the treatment of human and animal infections [38]. The ability of gentamicin to control 24 h old single species biofilms was analysed in terms of biomass removal (Figure 2 and Table 3). A biomass reduction was observed in all cases, with values between 59.3% (*S. aureus* Mannitol, MIC) and 85.7% (*P. aeruginosa* MH/BHI, 10× MIC). Interestingly, and although *P. aeruginosa* MH/BHI showed an intermediate susceptibility profile to gentamicin, the highest percentage of removal was observed for this isolate at the 10× MIC concentration. Comparing results between concentrations, and for the same isolate, the percentage of reduction increased significantly (*p* < 0.05) for all cases, except for *S. aureus* BPA1. For *S. aureus* BPA1, *S. aureus* BPA2, and *P. aeruginosa* MH/BHI, biomass removal was rated as high for both treatments. In the case of *S. aureus* Mannitol, for the MIC the effect was moderate (59.4%), while for 10× MIC it was high (78.1%).

Despite the percentages of removal being quite high, in none of the cases was there complete biofilm removal. This suggests that antibiotic treatment alone is not effective in removing the biofilm. The fact that the biofilm suffered a loss of biomass does not mean that it is not metabolically active and becomes unable to develop again [39]. In Van Laar et al. [39], biofilms of a clinical strain of *Klebsiella pneumoniae* isolated from a human wound were exposed to treatment with carbapenems (imipenem, meropenem and doripenem) at sublethal concentrations, and evaluated for their effect on the metabolism and gene expression of the biofilms. The authors observed that, despite the morphological differences, there was no significant difference in the viability of most of the treated biofilms. Furthermore, biofilms treated with all three antibiotics returned to pretreatment morphology shortly after antibiotic removal. Regarding gene expression, genes involved in peptidoglycan biosynthesis and catabolism have been implicated in these morphological transitions. Genes involved in the general stress response, virulence, and antibiotic resistance were differentially expressed in the presence of imipenem, leading to profound changes in cell physiology. Furthermore, upregulation of genes involved in the formation of persistent antibiotic-tolerant cells was observed for imipenem-treated biofilms, which possibly contributed to the ability of these biofilms to remain viable, even with a carbapenem treatment of 1000× MIC [39]. Indeed, the determination of the MIC, as well as the minimum bactericidal concentration (MBC), is based on planktonic cells, while the minimum biofilm eradication concentration (MBEC) is defined as the lowest concentration of antibiotic required to eradicate the biofilm [40]. However, the MBEC has yet to be introduced into a clinical setting/standardization, primarily due to inconsistent perception and use among clinicians and researchers [40]. In fact, the diagnosis of biofilms is still quite difficult, both in human and veterinary medicine [41]. Wounds infected by biofilms end up being treated as a simple infection, for which antibiotic treatment is the most common approach [42]. This results in a tolerance of the biofilm to the treatment, which causes the persistence of infections [43].

## 4. Conclusions

Undoubtedly, antimicrobial resistance (AMR) is a One Health issue, given the growing evidence that AMR can spread among animals, humans, and the environment [38]. This case report characterizes *S. aureus* and *P. aeruginosa* strains isolated from an equine chronic wound in terms of their susceptibility profile to antibiotics, evaluates their ability to form biofilms, and determines if a conventional antibiotic treatment would be able to control the formation of these biofilms. Overall, these initial findings suggest a polymicrobial wound infected by critical and high priority pathogens capable of forming biofilm. Although the potential for biofilm formation has been demonstrated in vitro, it was not possible to establish a correlation between the in vitro formation potential and the effective presence of biofilms in the wound in situ. However, this observation serves as an alert to the possibility of infection by biofilms, which may explain the chronicity of the wound. Furthermore, the fact that the wound was only disinfected with chlorhexidine may indicate a cross-resistance between biocides and antibiotics, or even a possible transmission of resistance.

## Figures and Tables

**Figure 1 animals-13-01342-f001:**
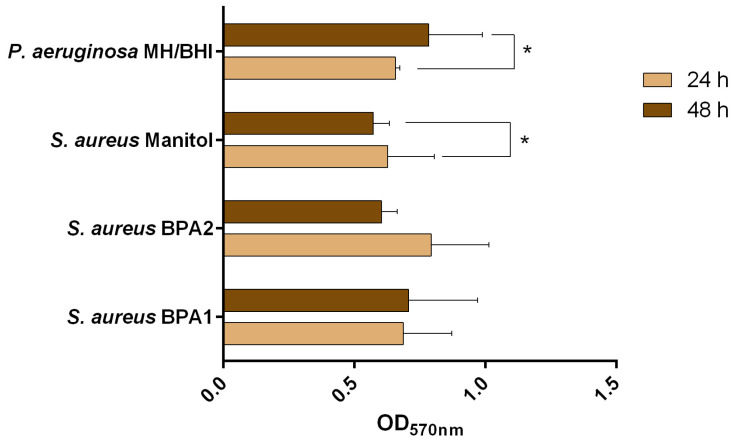
Values of optical density at 570 nm (OD_570nm_) as a measure of biomass of 24 h and 48 h old biofilms. The mean ± standard deviation of three independent experiments is illustrated. * *p* < 0.05.

**Figure 2 animals-13-01342-f002:**
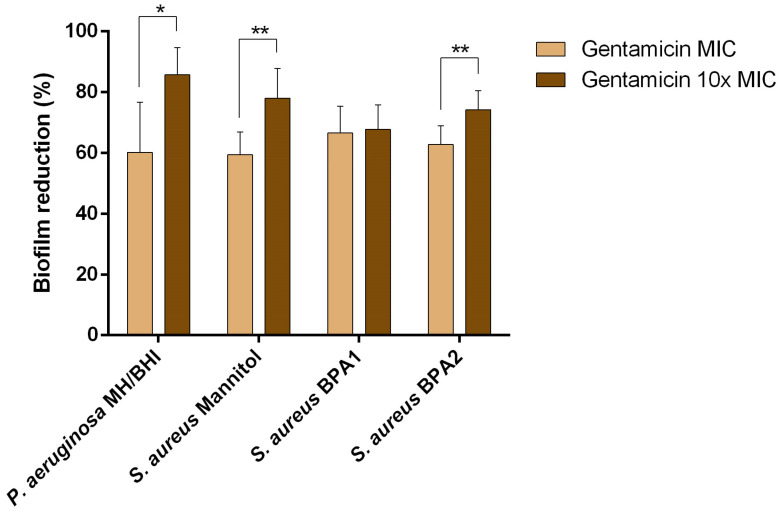
Percentage of biofilm reduction of *S. aureus* and *P. aeruginosa* isolates exposed to the MIC and 10× MIC of gentamicin for 24 h. The mean ± standard deviation of three independent experiments is illustrated. ** *p* ≤ 0.01; * *p* < 0.05.

**Table 1 animals-13-01342-t001:** Bacterial identification and antibiotic susceptibility profile using the VITEK^®^ 2 automatic system. Antibiotic susceptibility results were tested and read according to the recommendations of the Clinical and Laboratory Standards Institute (CLSI), and are expressed as sensitive (S), intermediate (I), or resistant (R).

	*Pseudomonas aeruginosa* (F09MH/BHI)	*Staphylococcus aureus* (F09Mannitol)	*Staphylococcus aureus* (BPA1)	*Staphylococcus aureus* (BPA2)
Ertapenem	**R** n.s.			
Imipenem	**R** MIC ≥ 16 µg/mL			
Amikacin	**S** MIC = 4 µg/mL			
Gentamicin	**I** MIC = 4 µg/mL	**S** MIC ≤ 0.5 µg/mL	**S** MIC ≤ 0.5 µg/mL	**S** MIC ≤ 0.5 µg/mL
Enrofloxacin	**R** MIC ≥ 4 µg/mL	**S** MIC ≤ 0.5 µg/mL	**S** MIC ≤ 0.5 µg/mL	**S** MIC ≤ 0.5 µg/mL
Marbofloxacin	**R** MIC ≥ 4 µg/mL	**S** MIC ≤ 0.5 µg/mL	**S** MIC ≤ 0.5 µg/mL	**S** MIC ≤ 0.5 µg/mL
Kanamycin		**S** MIC ≤ 4 µg/mL	**S** MIC ≤ 4 µg/mL	**S** MIC ≤ 4 µg/mL
Neomycin		**S** MIC ≤ 2 µg/mL	**S** MIC ≤ 2 µg/mL	**S** MIC ≤ 2 µg/mL
Pradofloxacin		**S** MIC ≤ 0.12 µg/mL	**S** MIC ≤ 0.12 µg/mL	**S** MIC ≤ 0.12 µg/mL
Erythromycin		**S** MIC ≤ 0.25 µg/mL	**S** MIC ≤ 0.25 µg/mL	**S** MIC ≤ 0.25 µg/mL
Clindamycin		**S** MIC = 0.25 µg/mL	**S** MIC = 0.25 µg/mL	**S** MIC = 0.25 µg/mL
Doxycycline		**S** MIC ≤ 0.5 µg/mL	**S** MIC ≤ 0.5 µg/mL	**S** MIC ≤ 0.5 µg/mL
Tetracycline		**S** MIC ≤ 1 µg/mL	**S** MIC ≤ 1 µg/mL	**S** MIC ≤ 1 µg/mL
Nitrofurantoin		**S** MIC ≤ 16 µg/mL	**S** MIC ≤ 16 µg/mL	**S** MIC ≤ 16 µg/mL
Chloramphenicol		**S** MIC ≤ 4 µg/mL	**S** MIC ≤ 4 µg/mL	**S** MIC ≤ 4 µg/mL
Trimethoprim/sulfamethoxazole		**S** MIC ≤ 10 µg/mL	**S** MIC ≤ 10 µg/mL	**S** MIC ≤ 10 µg/mL
Cefoxitin screening test		*Negative*	*Negative*	*Positive*
Oxacillin		n.s. MIC = 1 µg/mL	n.s. MIC = 0.25 µg/mL	n.s. MIC ≤ 0.25 µg/mL

n.s.—not specified.

**Table 2 animals-13-01342-t002:** Biofilm formation classification according to Stepanović et al. [17]. For moderate biofilm producers: 2 × ODc < OD ≤ 4 × ODc; for strong biofilm producers: OD > 4 × ODc. The considered ODc value was 0.193 ± 0.030.

	*Pseudomonas aeruginosa* (F09MH/BHI)	*Staphylococcus aureus* (F09Mannitol)	*Staphylococcus aureus* (BPA1)	*Staphylococcus aureus* (BPA2)
**24 h**	Moderate	Moderate	Moderate	Moderate
**48 h**	Strong	Moderate	Moderate	Moderate

**Table 3 animals-13-01342-t003:** Classification of treatment efficacy in biofilm removal according to Lemos et al. [34]. For moderate efficacy: ≥25% to <60%; for high efficacy: ≥60% to <90%.

	*Pseudomonas aeruginosa* (F09MH/BHI)	*Staphylococcus aureus* (F09Mannitol)	*Staphylococcus aureus* (BPA1)	*Staphylococcus aureus* (BPA2)
**MIC**	High	Moderate	High	High
**10** × **MIC**	High	High	High	High

## Data Availability

All raw data are available from the corresponding authors upon reasonable request.

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
