# Peer review of "Biofilm Production by Critical Antibiotic-Resistant Pathogens from an Equine Wound"

_animals, 2023, doi:10.3390/ani13081342_

Round 1
Reviewer 1 Report
In the article, the authors described the drug susceptibility profiles and biofilm production of Pseudomonas aeruginosa and Staphylococcus aureus strains which might be the pathogens of an equine wound. Overall, the manuscript is well written and clear in its results. But the study’s design and conclusion that Pseudomonas aeruginosa and Staphylococcus aureus strain had the ability to form biofilms and could transfer between animals, humans and environment had been reported by many other researchers. So, I suggest the study publish as short report rather than original article.
Other suggestions and points to the manuscript are listed below:
1、Line 40. “Samples were cultured in Brain Hearth Infusion (BHI, Oxoid, Hampshire, UK) broth 89 and incubated at 37 ± 2 °C, for 24 h. After this period, inocula with turbidity were consid-90 ered positive for bacterial growth and selective and differential growth mediums [GSP 91 Pseudomonas Aeromonas Selective Agar (Merck, Taufkirchen, Germany), Chromocult Col-92 iform Agar (Oxoid, Hampshire, UK), MacConkey (Oxoid, Hampshire, UK), Baird-Parker 93 Agar (Oxoid, Hampshire, UK) and Manitol (Oxoid, Hampshire, UK)] were used for the 94 isolation process”
Samples cultured in BHI broth for 24h would make the strains with low nutritional requirements grow preferentially, thus inhibiting the growth of other strains with high nutritional requirements. Why not culture the samples on different growth medium agar plate firstly?
2、 Line 124. “ sterile NaCl” with what concentration?
3、 Line110. “ 3.3. Minimum inhibitory concentration (MIC) determination” shoud be 2.3
4、 Line 146. “The remaining attached cells were treated as referred to at 2.3.1”, there were no 2.3.1
5、 Line 164.” As described at section 3.1.,” there were no section 3.1,please check the section numbers and correct.
Author Response
The authors carefully revised the manuscript considering all the suggestions and comments proposed by the reviewers. All the required modifications were performed. The English language and all the sections have been improved and supplemented with more relevant information (considering the indications of the reviewers). We found the comments and suggestions very useful to improve the quality of the study and make it clearer.
We revised using the following approach:
- Reformulation of the manuscript when needed, using the “Track Changes” function of MS Word. We recommend viewing “Track Changes” with the option “Show All Revisions Inline” so that the line numbering shown below is correct.
- Our answers to the comments and recommendations proposed according “Example for author to respond reviewer-MDPI”
Reviewer #1
In the article, the authors described the drug susceptibility profiles and biofilm production of Pseudomonas aeruginosa and Staphylococcus aureus strains which might be the pathogens of an equine wound. Overall, the manuscript is well written and clear in its results. But the study’s design and conclusion that Pseudomonas aeruginosa and Staphylococcus aureus strain had the ability to form biofilms and could transfer between animals, humans and environment had been reported by many other researchers. So, I suggest the study publish as short report rather than original article.
Response: We acknowledge the Reviewer for this comment and are glad that the reviewer found the article to be clear well-written. We agree with the Reviewer's suggestion that the study fits more into a Case Report. Changes throughout the entire manuscript have been made accordingly.
Other suggestions and points to the manuscript are listed below:
- 1. Line 40. “Samples were cultured in Brain Hearth Infusion (BHI, Oxoid, Hampshire, UK) broth and incubated at 37 ± 2 °C, for 24 h. After this period, inocula with turbidity were considered positive for bacterial growth and selective and differential growth mediums [GSP Pseudomonas Aeromonas Selective Agar (Merck, Taufkirchen, Germany), Chromocult Coliform Agar (Oxoid, Hampshire, UK), MacConkey (Oxoid, Hampshire, UK), Baird-Parker Agar (Oxoid, Hampshire, UK) and Mannitol (Oxoid, Hampshire, UK)] were used for the isolation process”
Samples cultured in BHI broth for 24h would make the strains with low nutritional requirements grow preferentially, thus inhibiting the growth of other strains with high nutritional requirements. Why not culture the samples on different growth medium agar plate firstly?
Response 1: We are grateful to the Reviewer for spotting this. BHI medium was selected because it is the routine medium used in the laboratory. In fact, according to the Handbook of Media for Clinical Microbiology, BHI medium is a nutrient-rich medium suitable for the growth of many species of microorganisms.
Snyder, J.W., & Atlas, R.M. (2006). Handbook of Media for Clinical Microbiology (2nd ed.). CRC Press. https://doi.org/10.1201/9781420005462
- Line 124. “ sterile NaCl” with what concentration?
Response 2: We thank the reviewer for spotting our omission. This information has been added (L. 132).
- Line110. “ 3.3. Minimum inhibitory concentration (MIC) determination” should be 2.3
Response 3: We agree and was changed accordingly. It is now corrected (L. 116).
- 4. Line 146. “The remaining attached cells were treated as referred to at 2.3.1”, there were no 2.3.1
Response 4: We thank the reviewer for spotting this typo and it is now corrected. Section 2.3.1. became 2.4 (L.154).
- 5. Line 164.” As described at section 3.1.,” there were no section 3.1, please check the section numbers and correct.
Response 5: We agree with the reviewer and we changed this section. We referred to section 2.1 (L. 175). The numbering of all sections has been checked.

Reviewer 2 Report
The manuscript describes the isolation of relevant bacterial pathogenes from a chronic wound of an equine. This is an interesting subject, but the study should be presented as a case report instead of an article. Some modifications are required, as well as revision of English language.
1 - Simple summary, line 22 - The elements of the One Health triad are all interconnected. This sentence must be corrected.
2 - Abstract line 32, and throughout the text - As the clonal characterization of the isolates was not performed, the term strains must be replaced by the term isolates throughout all the manuscript text.
3 - Material and methods, lines 116-133 - Remove subtitle 2.4.1., it is not necessary. The biofilm formation protocol must be better described, as some details are not clear. For example, what was the incubation period for biofilm formation?
4 - Material and methods, lines 134-156 - Remove subtitle 2.5.1. In this point, it is not clear what OD corresponds to the OD untreated 570nm. Is it the OD of the positive control?
5 - Discussion, line 187-191 - The importance of the EMA classes of antibiotics should be better discussed.
6 - Discussion, lines 205-206 - Explain this sentence, as it is not according to EUCAST guidelines, which consider intermediate susceptibility as susceptible, high-dose.
7 - Discussion - Although the wound infection was not being treated with antibiotics, it was subjectic to chlorhexidine. As such, authors should discuss the possibility of cross-resistance between biocides and antibiotics.
Minor issues:
- line 27 - substitute ; by ,
- line 36 - ... neither to the mare...
- line 40 - ... for the P. aeruginosa isolate (at ...
- line 43 - substitute the word phenomenon by possibility
- lines 75-76 - ... exposed to gentamicin, and the effectiveness of this antimicrobial in biofilm removal was assessed.
- line 196 - recommenda-tions
- line 209 - S. aureus is not in italics
- line 217 vs line 230 - 24 h vs 24h - uniformize throughout the manuscript
- lines 305 and 306 - in vitro is not in italics
Author Response
The authors carefully revised the manuscript considering all the suggestions and comments proposed by the reviewers. All the required modifications were performed. The English language and all the sections have been improved and supplemented with more relevant information (considering the indications of the reviewers). We found the comments and suggestions very useful to improve the quality of the study and make it clearer.
We revised using the following approach:
(i) Reformulation of the manuscript when needed, using the “Track Changes” function of MS Word. We recommend viewing “Track Changes” with the option “Show All Revisions Inline” so that the line numbering shown below is correct.
(ii) Our answers to the comments and recommendations proposed according “Example for author to respond reviewer-MDPI”
Reviewer #2
The manuscript describes the isolation of relevant bacterial pathogens from a chronic wound of an equine. This is an interesting subject, but the study should be presented as a case report instead of an article. Some modifications are required, as well as revision of English language.
- We acknowledge the Reviewer for this comment and for finding the subject interesting. We agree with the Reviewer's suggestion that the article fits more into a Case Report. Changes throughout the entire manuscript have been made accordingly. English revisions have been made throughout the text.
1 - Simple summary, line 22 - The elements of the One Health triad are all interconnected. This sentence must be corrected.
Response 1: We thank the Reviewer for highlighting this sentence. The sentence has been corrected (L. 22).
2 - Abstract line 32, and throughout the text - As the clonal characterization of the isolates was not performed, the term strains must be replaced by the term isolates throughout all the manuscript text.
Response 2: We thank the Reviewer for this comment. The term has been replaced throughout the entire manuscript.
3 - Material and methods, lines 116-133 - Remove subtitle 2.4.1., it is not necessary. The biofilm formation protocol must be better described, as some details are not clear. For example, what was the incubation period for biofilm formation?
Response 3: We thank the reviewer for this suggestion and agree that the biofilm formation protocol must be better described. This method has been rewritten with more detailed information (L. 124-143). We changed this section to make it clearer to the reader. The subtitle has also been removed.
4 - Material and methods, lines 134-156 - Remove subtitle 2.5.1. In this point, it is not clear what OD corresponds to the OD untreated 570nm. Is it the OD of the positive control?
Response 4: We thank the reviewer for this suggestion. The subtitle has been removed and it was clarified what the OD and the OD untreated correspond to (L. 139-141; L. 162).
5 - Discussion, line 187-191 - The importance of the EMA classes of antibiotics should be better discussed.
Response 5: We thank the reviewer for this suggestion. We have taken the Reviewer's suggestion into account and this part of the discussion has been clarified (L. 208-219).
6 - Discussion, lines 205-206 - Explain this sentence, as it is not according to EUCAST guidelines, which consider intermediate susceptibility as susceptible, high-dose.
Response 6: We thank the reviewer for this comment. However, in the lines the Reviewer refers to (now L. 233-234) no mention is made to EUCAST guidelines. We believe the Reviewer was referring to lines 233-234, where the P. aeruginosa isolate is stated to show an intermediate profile of resistance to gentamicin. If so, this classification as "intermediate" is correct because the antibiogram was performed and interpreted according to CLSI and not EUCAST guidelines. To avoid misinterpretations, all references to EUCAST throughout the manuscript have been removed (L. 199-202).
7 - Discussion - Although the wound infection was not being treated with antibiotics, it was subjectic to chlorhexidine. As such, authors should discuss the possibility of cross-resistance between biocides and antibiotics.
Response 7: We thank the reviewer for this great suggestion. In fact, several studies and with different strains have shown that subinhibitory concentrations of chlorhexidine induce resistance to chlorhexidine and decrease susceptibility to antibiotics. This information was added in the manuscript (L. 239-242; L. 35-36).
Minor issues:
- line 27 - substitute ; by ,
- line 36 - ... neither to the mare...
- line 40 - ... for the P. aeruginosa isolate (at ...
- line 43 - substitute the word phenomenon by possibility
- lines 75-76 - ... exposed to gentamicin, and the effectiveness of this antimicrobial in biofilm removal was assessed.
- line 196 - recommendations
- line 209 - S. aureus is not in italics
- line 217 vs line 230 - 24 h vs 24h - uniformize throughout the manuscript
- lines 305 and 306 - in vitro is not in italics
Response: We thank the reviewer for spotting all these minor issues. All are now corrected (L. 27; L. 36; L. 40; L. 44; L. 77-78; L. 193; L. 206; L. 302-303).

Reviewer 3 Report
The manuscript reveals highlights the phenomenon of resistance transmission between animals, animals and humans or animals and the environment.
The design of this paper is well but the number of strains tested is too small. This cannot be representative of the experiment, and it is a serious experimental error. Moreover, even though the sample is a equine-derived pathogen, the supporting references are mostly for humans, and the interpretation of the experimental results has no particular meaning for the equine. Therefore, please increase the number of samples and emphasize the importance of words in sentences.
Below are general comments:
1. Line 53.
I understand ‘the microbiology of equine wounds and biofilm infection has been limitedly studied’. But you should write down the biofilm infection of other animals because there is a clear difference between humans and animals in biofilm infection.
2. Line 110.
You should clearly list all the antibiotics used in the MIC.
3. Line 169.
Reference 22 is about human, not the animals or equine. You have to find reference about animals
4. Line 173
In this study, S. aureus and P. aeruginosa have resistance about methicillin, vancomycin and carbapenem, respectively? If not, it is not appropriate to include this sentence.
5. Line 288
I saw antibiotic resistance and biofilm in equine, but there is no content on the characteristic meaning and implications of equine.
Author Response
The authors carefully revised the manuscript considering all the suggestions and comments proposed by the reviewers. All the required modifications were performed. The English language and all the sections have been improved and supplemented with more relevant information (considering the indications of the reviewers). We found the comments and suggestions very useful to improve the quality of the study and make it clearer.
We revised using the following approach:
(i) Reformulation of the manuscript when needed, using the “Track Changes” function of MS Word. We recommend viewing “Track Changes” with the option “Show All Revisions Inline” so that the line numbering shown below is correct.
(ii) Our answers to the comments and recommendations proposed according “Example for author to respond reviewer-MDPI”
Reviewer #3
The manuscript reveals highlights the phenomenon of resistance transmission between animals, animals and humans or animals and the environment.
The design of this paper is well but the number of strains tested is too small. This cannot be representative of the experiment, and it is a serious experimental error. Moreover, even though the sample is a equine-derived pathogen, the supporting references are mostly for humans, and the interpretation of the experimental results has no particular meaning for the equine. Therefore, please increase the number of samples and emphasize the importance of words in sentences.
R: We acknowledge the Reviewer for this comment. In fact, there was a flaw regarding the limited use of references with works with animals. New and appropriate references have been added to the manuscript. Regarding the Reviewer's concern about the number of strains tested, we do not consider this to be an experimental error. We believe that only these strains were isolated because, as observed in the study, they are strains with associated resistance. Since this animal was being medically monitored and the wound being treated with an antiseptic, it is expected that chlorhexidine susceptible strains were not isolated. Moreover, in the study by Freeman et al. (2009) the authors showed that a number of identified bacteria were viable in the wound but nonculturable.
Freeman K, Woods E, Welsby S, Percival SL, Cochrane CA. Biofilm evidence and the microbial diversity of horse wounds. Can J Microbiol. 2009 Feb;55(2):197-202. doi: 10.1139/w08-115. PMID: 19295652.
Below are general comments:
- Line 53. I understand ‘the microbiology of equine wounds and biofilm infection has been limitedly studied’. But you should write down the biofilm infection of other animals because there is a clear difference between humans and animals in biofilm infection.
Response 1: We thank the Reviewer for this comment. New information and references on biofilms and wounds in animals have been added to the manuscript (L. 50-55; L. 180-181; L. 184-186).
- Line 110. You should clearly list all the antibiotics used in the MIC.
Response 2: We thank the Reviewer for this suggestion. Considering that this information is already described in Table 1 and taking into account the number of words for a Case Report, a sentence was added to direct to Table 1 (L. 121).
- Line 169. Reference 22 is about human, not the animals or equine. You have to find reference about animals
Response 3: We completely agree, and it was corrected accordingly. This and other references have been replaced by works on animals (L. 180-181; 184-186).
- Line 173. In this study, S. aureus and P. aeruginosa have resistance about methicillin, vancomycin and carbapenem, respectively? If not, it is not appropriate to include this sentence.
Response 4: We thank the Reviewer for highlighting this sentence. In fact, we were referring to the WHO priority list which classifies S. aureus as a high priority pathogen when it is methicillin resistant and resistant (or with an intermediate profile) to vancomycin. To avoid misinterpretation, this information has been restated (L. 186-189).
- Line 288. I saw antibiotic resistance and biofilm in equine, but there is no content on the characteristic meaning and implications of equine.
Response 5: We thank the Reviewer for this suggestion. Information about the meaning and complications for the animal has been added (L. 50-56).

Round 2
Reviewer 1 Report
The manuscript have been carefully revised considering all the suggestions and comments.
Author Response
Author's Reply to the Review Report (Reviewer 1)_Round 2
The authors carefully revised the manuscript considering all the suggestions and comments proposed by the reviewer 1. All the required modifications were performed. The English language and all the sections have been improved and supplemented with more relevant information (considering the indications of the reviewers). We found the comments and suggestions very useful to improve the quality of the study and make it clearer.
We revised using the following approach:
- Reformulation of the manuscript when needed (with the modifications highlighted in yellow).
- Our answers to the comments and recommendations proposed.
Reviewer #1
The manuscript have been carefully revised considering all the suggestions and comments.
R: We acknowledge the Reviewer for all the suggestions and comments. Minor spelling corrections have been made throughout the manuscript.

Reviewer 2 Report
The manuscript version submitted in the platform does not include authors' corrections as stated in the author's reply file (see as example answer to comments 1 and 2), so I request that the authors submit the last version of the manuscript (with the changes marked in track changes or highlighted in yellow) so I can make my final decision.
Author Response
Author's Reply to the Review Report (Reviewer 2)_Round 2 and Round 3
The authors carefully revised the manuscript considering all the suggestions and comments proposed by the reviewer #2. All the required modifications were performed. The English language and all the sections have been improved and supplemented with more relevant information (considering the indications of the reviewers). We found the comments and suggestions very useful to improve the quality of the study and make it clearer.
We revised using the following approach:
(i) Reformulation of the manuscript when needed (with the modifications highlighted in yellow).
(ii) Our answers to the comments and recommendations proposed.
Reviewer #2
The authors have addressed all my comments except for one, but I guess my remark was not clear enough.
In line 208-209, the authors state that "Furthermore, it has an intermediate susceptibility profile to the aminoglycoside gentamicin, which suggests that it is on the verge of acquiring resistance to this antibiotic."
Please explain how you can conclude that an isolate that presents an intermediate susceptibility profile "is on the verge of acquiring resistance to this antibiotic".
R: We thank the Reviewer for highlighting this sentence. The sentence has been reformulated in order to avoid misunderstandings (L. 201-203). Moreover, minor spelling corrections have been made throughout the manuscript. Changes are highlighted in yellow.

Round 3
Reviewer 2 Report
The authors have addressed all my comments except for one, but I guess my remark was not clear enough.
In line 208-209, the authors state that "Furthermore, it has an intermediate susceptibility profile to the aminoglycoside gentamicin, which suggests that it is on the verge of acquiring resistance to this antibiotic."
Please explain how you can conclude that an isolate that presents an intermediate susceptibility profile "is on the verge of acquiring resistance to this antibiotic".
Author Response

(The authors gave the same response as above.)
